# Work-family conflict among hotel housekeepers in the Balearic Islands (Spain)

Xenia Chela-Alvarez[ID][1,2,3]*, M. Esther Garcia-Buades[4], Victoria A. Ferrer-Perez[ID][4], Oana Bulilete[1,2,3], Joan Llobera[1,2,3]

1 Primary Care Research Unit of Mallorca, Balearic Islands Health Services, Palma, Spain, 2 GrAPP-caIB– Health Research Institute of the Balearic Islands (IdISBa), Palma, Spain, 3 RICAPPS- Red de Investigación Cooperativa de Atención Primaria y Promoción de la Salud–Carlos III Health Institute (ISCIII), Madrid, Spain, 4 Department of Psychology, University of the Balearic Islands, Palma, Spain

* xenia.chela@ibsalut.es

## Abstract

The massive incorporation of women to the labour market has increased academic and applied interest on work-life issues throughout the years. This article aims to describe the domestic burden and difficulties in work-life balance (WLB) and to understand the intersection of work and family spheres among hotel housekeepers (HHs). A cross-sectional study was conducted through Primary Health Care in the Balearic Islands (Spain); 1,043 HHs were enrolled. 56.7% reported difficulties in WLB. Risk factors for perceiving difficulties in WLB were: living with someone else (regardless of the number of co-habitants), having difficulties making ends meet, being the main person in charge of domestic tasks, having a dependant, having an external locus of control, presenting higher levels of stress at work, working more hours a week and being younger. Protective factors from experiencing work-family conflict (WFC) were job and wage satisfaction. WFC is strongly influenced by individual, economic, labour and domestic factors: these relationships show that labour and domestic spheres are non-separate worlds.

## Introduction

The massive incorporation of women to the labour market in the past decades raised the interest on the articulation of paid and unpaid work. Western societies have been traditionally organized around the sexual division of labour, by which men have been the main responsible partner for paid work and women for unpaid care work [1]. This division of labour has been challenged by the growth in the number of women entering the labour market [2].

The term "work-life balance" (WLB) can be defined as the ability to successfully manage different demands coming from different life domains (i.e. allocating enough time and resources, being satisfied and involved in a balanced manner across life domains) [3]. Life domains can be understood as work-related (including just paid work) and non-work related (the rest of activities: leisure, caring, learning, etc). Since demands across life domains vary during life course, WLB acquires different meanings depending on the situation of the person [4].The importance of the concept raised during the 1960s, when a large amount of women

**Funding:** This research is part of a wider Project, "Hotel Housekeepers and Health" (PI: JLL), which is funded by Sustainable Tourism's Tax Fund (Balearic Islands Government), grant number ITS-17-096. The funders had no role in study design, data collection and analysis, decision to publish, or preparation of the manuscript.

**Competing interests:** The authors have declared that no competing interests exist.

entered the formal labour market, while remaining in charge for domestic and reproductive tasks [5]; this situation entailed a challenge to the traditional sexual division of labour [2]. Since then, WLB has been a main political and social issue related to gender (in)equality [6]; WLB of working parents and carers has been the focus of several studies (see e.g.Rapoport & Rapoport, 1965 [7]), reports and laws [8–10] in recent decades.

Paid and unpaid work run with different logics that might result in incompatible demands. From a role-conflict approach, role responsibilities at work generate negative consequences on non-work roles and vice versa, in the so-called "work-family conflict" (WFC) [11, 12]. Conflict may arise in two directions: work demands may interfere with family demands or family demands may interfere with work demands, although both conflict directions are moderately correlated [13].Throughout the article, the term WFC refers to conflict in general, that is, to what extent the responsibilities of one life sphere interfere with the accomplishment of the responsibilities in other life domains regardless of the direction in which it takes place [11]. Hence, the absence of work-life balance implies conflict.

When balance between work and non-work life is achieved, several positive outcomes have been described at different levels. Positive outcomes of WLB at work have been reported such as increases in job performance and satisfaction, organizational commitment, and decreases in job burnout and absenteeism [12, 14]. Negative outcomes of WFC include a reduction of life satisfaction [15], a raise in poor health condition, cognitive problems and conflicts with family members [12]. Additionally, as WFC increases, psychological distress, hypertension and incidence of alcohol abuse, among others, increase as well [12]. Due to the importance of the work and non-work interface for the individual well-being, WFC is a significant research topic in studies considering work and quality-of-life.

Jobs in the hospitality sector have been associated to a greater job stress [16]. In the Balearic Islands (Spain), where the tourist sector contributed to 44% GDP (pre-pandemic data) [17], hotel housekeepers (HHs) represent a very important occupational group (approx. 13,000). HHs' job is characterized by low control and high physical demands [18, 19], which may entail not having enough time off to recover from the job demands faced on daily basis. In the Balearic Islands, although most HHs work 40 hours per week, in a continuous working-day and have little options for flexible scheduling, they are assigned a fixed amount of rooms that must be cleaned every day. In addition, HHs work more intensely during the Summer season, when schools are closed. Given the insufficient public-funded children-care services, the mostly private children-care services available are non-affordable for many families. Therefore, we can expect HHs being an occupational group presenting WFC. In this context, the objectives of this study are to describe the domestic burden of HHs in the Balearic Islands, to assess the perception about difficulties in balancing work and non-work life and to understand the factors that explain WFC.

## Antecedents of WFC

Several studies have focused on the antecedents of WFC, identifying individual, sociodemographic, family and domestic, and work factors.

Locus of control (LOC) has been identified as an antecedent of WFC at the individual level. LOC refers to a personality trait understood as the perception a person has about the control of the events that occur in their life [20]. LOC can be internal -those who think that things that happen in their life are the consequence of their attitudes, actions and behaviour- or external -those who attribute what happens in their life to fortune, fate or other's decisions-[20].Previous evidence found that people with internal LOC tend to experience low levels of WFC [21–23]; therefore, we can expect that HHs with an external LOC will have higher levels of WFC.

Regarding family factors, being a parent of younger children and the number of children have been reported as making WLB more difficult [6, 11]. This is because demands from work settings make it more difficult to reconcile with those from the domestic sphere and vice versa.

Labour/work variables described as making WLB more difficult are the following: having a temporary contract [6], higher number of working hours [6, 11, 12, 24], inflexibility of work schedule [11], time pressure [12] and high job demands [12].On the contrary, a continued working schedule was identified by HHs in the Balearic Islands as a facilitator for combining work and family responsibilities [19].

Social support has been studied as a contextual antecedent of WFC and has been defined as "the degree to which a person's basic social needs are gratified through interaction with others. Basic social needs include affection, esteem or approval, belonging, identity, and security" [25; p.147]. Studies have focused on the relation between work-family support and WFC, showing that individuals presenting higher levels of social support report lower levels of work-family conflict [23, 26, 27]. Additionally, Haines et al. (2019) [28] found that women reported more social support outside work, which was associated with lower WFC. We expect that HHs with lower social support will present more difficulties balancing work and family spheres.

Additionally, some structural factors such as gender and social class inequalities have an impact on WFC. In Spain, despite the growth of the participation of women in paid work/labour market, they tend to remain linked to care and domestic work, while men remain doing a small proportion of unpaid work [29–31]. The Spanish Time-Use Survey (2009–10) indicated that women in the Balearic Islands spent 3 hours and 59 minutes daily in households and family activities, while men spent 2 hours and 23 minutes [32]. In Spain, data for 2010 from the European Institute of Gender Equality (EIGE) reported that women spent 2 hours and 57 minutes per day in household and family care, and men 1 hour and 18 minutes [33]. This represents a gender gap, i.e. a "disproportionate difference between men and women and boys and girls, particularly as reflected in attainment of development goals, access to resources and levels of participation" [34]. This gender gap has consequences in other areas of life and is influenced by gender norms [35, 36]. Despite this unequal contribution, there is mixed evidence whether women and men report different degrees of WFC [6, 28, 30, 37].

Social class also plays a central role in WLB. Evidence on the association of social class, gender and the division of domestic and care tasks pointed out to the fact that working class women appear to provide the bulk of social care [38]. Further, time women devote to housework varies depending on their relative income, so that as income increases, time devoted to housework decreases. This inverse relationship seems to be present up to the point at which women contribute to half the household income [29]. However, Altuzarra et al. (2020) [29] reported that income was not associated with time devoted to childcare. Widely, working class people have been reported as those with higher levels of imbalance between their work and life spheres [6, 39] and higher household income has been related to lower WFC [28]. Despite this, most studies on WFC have not focused on working class or unskilled workers [39]. Given the fact that HHs are mostly women and considered unskilled manual workers, we expect relevant levels of WFC.

WLB is also framed by institutional context, such as the welfare state regime. Spain has been labelled as a Southern welfare state regime along with Italy, Greece and Portugal-, characterized by low levels of social expenditure; higher proportion of the population at risk of poverty; higher inequality levels; the presence of strong 'familism', a term to describe that family (women) has a central role in providing care work [38, 40]; and the underdevelopment of family benefits and services [38, 41]. Spain's expenditure on family and children was 0.9% of GDP in 2019, while the EU-27 average was 1.8% [42]. In face of that, informal caregiving provided by families (mainly, by women) is a frequent strategy to attend care needs and reconcile work

and family life [41, 43] and women become the main caregivers [44]. Furthermore, research has explored the importance of the cultural context as significant for the work-family interface [45].

The objectives of this study are to describe the domestic burden of HHs in the Balearic Islands, focusing on the characteristics of the household tasks HHs carried out; and to assess the perception about their difficulties in balancing work and non-work life. Finally, with the aim to understand the intersection of work and family spheres, we explore which factors related to individual, work and domestic spheres explain the assessment of easiness and difficulties in balancing work and family life. Considering that most studies on WFC have not focused on working class or unskilled workers [39], and the importance of the intersection of gender and social class, this study contributes to filling the gap on WFC and working class in the literature.

## Materials & methods

This is a cross-sectional study, conducted at centres of Primary Health Care in the Balearic Islands between November 2018 and February 2019. The study was carried out in 39 out of 58 primary health care centres in the Balearic Islands. It is noteworthy that data collection was made previous to the COVID19 pandemic.

### Subjects

HHs who were older than 18 years, had health coverage in the Balearic Public Health System, worked as HHs during 2018, and were willing to participate in this study were included after signing the informed consent. Those who presented a language barrier were excluded.

An initial list of about 13,000 possible HHs was available from the Balearic Public Health Services. A randomly selected sample of 1,043HHs was needed to estimate population parameters, with a 3% precision, and 95% confidence.

### Data collection

Before inclusion, nurses were trained for conducting the study. They were in charge of recruiting HHs. The nurses called the HHs by telephone, asked for inclusion criteria, and set up a date for the visit. All study visits were carried out in the Primary Care Health Centres and the interviews lasted for one hour approximately.

In order to enrol the amount of HHs needed, 4,436 phone calls were made (some HHs were called more than once in order to be contacted or recruited). Finally, 1,043 HHs were enrolled: 773 in Mallorca, 89 in Menorca, 137 in Ibiza and 44 in Formentera. All participants were female.

### Measures

**Dependent variables.** WFC was measured through the difficulties reported in WLB in response to the question "How easy or difficult is it for you to combine work with your care responsibilities?". The respondent gave an answer of "very easy", "rather easy", "rather difficult", "very difficult", "do not know/do not answer". In the bivariate analysis "very easy" and "rather easy" were categorized as "easy", and "rather difficult" and "difficult" as "difficult". This question was taken from the European Quality of Life Survey (EQLS, 2016) [6].

**Independent variables.**  *(1) Individual and sociodemographic.*

- We assessed locus of control (LOC) through the question "To what extent do you agree with the following sentence:"I feel what happens in my life is often determined by factors

beyond my control", with Likert-type response options ranging from 1 ("completely disagree") to 6 ("completely agree"). This question was validated in English by Mounce et al. (2018) [46]; a forward translation into Spanish was made as recommended by Ortiz-Gutiérrez et al. (2018) [47]. Internality is considered when the respondent answers the options 1 to 3 (disagreement statements).

- Frequency of being overwhelmed by daily tasks was asked by "How often would you say you feel overwhelmed by tasks you have to perform usually?"; with 3 response options ("very often", "sometimes", "rarely"). This question was taken from the Spanish Time Use Survey (2002–2003) [48].

- Socio-demographic questions asked about age, nationality, and educational level.

*(2) Domestic and familiar.*

- Number of co-habitants. First, HHs were asked whether they were living alone. In case they did not live alone, HHs were asked who were they living with. This variable was operationalized as number of people living in the household.

- The distribution of domestic tasks and care of people were evaluated through the question "At home, who is mainly in charge of household chores and the care of people who are not able of caring for themselves?". This question had to be answered in relation to four items -drawn from the Spanish Health Survey 2011–12 [49]-: taking care of people younger than 15 years old, taking care of a disabled person, taking care of people older than 74 years old, and responsibility for domestic tasks (cleaning, ironing, cooking,...). For each item, the following response options were available: they alone; their partner alone; shared with their partner; shared with another person who was not their partner; another person from home who was not their partner; a hired person; a person who does not reside in the house; social services; him/herself; another situation; do not know/do not answer. It was a multiple answer question. In order to simplify the analysis and the results, for bivariate and multivariate analyses we transformed the care of dependents into a dichotomous variable: having or not a dependant (including the three categories). The variable "main person in charge of domestic tasks" was dichotomized as well: being the main person in charge vs. other situation.

- Social support was measured by DUKE-UNC-11 [50], an 11-item questionnaire that measured functional elements of social support (including confidant and affective support) and validated for the Spanish population (Cronbach's Alpha = 0.90) [51, 52]. Each item is valued in a 5-points scale ranging from 1 ("much less than I would like") to 5 points ("as much as I would like"). A final score ranging from 5 to 55 is obtained; 32 points or below correspond to a low social support and more than 32 points correspond to a normal social support [52].

- Difficulties making ends meet was measured by asking "A household might have different sources of income and more than a member can contribute with his/her income. Thinking about the total of your household monthly income, can you make ends meet. . .?" with 6 response options: "very easy", "easy", "rather easy", "a bit difficult", "very difficult", "do not know/do not answer". The question was taken from the EQLS 2016 [6].

*(3) Labour.*

- The instrument included questions on type of contract (permanent, recurring-seasonal or temporary), type of accommodation (apartment, hotel, etc.), hotel category, number of

rooms cleaned per day, years working as HHs, months worked last tourist season, and hours worked per week.

- Stress perceived at work was measured with the question "Globally and considering the conditions under which you do your work, indicate your level of stress at work in a scale from 1 ("very stressful") to 7 ("not at all stressful")".

- Job and wage satisfaction were assessed with the following questions: "Taking into account the characteristics of your job, indicate to what extent you consider your job satisfactory", with a response scale from 1 ("not satisfactory at all") to 7 ("very satisfactory"), and "To what extent are you satisfied with the salary you receive?", with a response scale from 1 ("not satisfied at all") to 7 ("very satisfied").

### Statistical analysis

Categorical variables (such as nationality, educational level, type of contract. . .) are presented in absolute numbers along with percentages, while quantitative variables (years working as HHs, months worked/year, level of stress,. . .) are presented as means and standard deviations (SD).

To assess the relationship between sociodemographic and labour variables with difficulties in WLB the chi-squared test and Student's t-test were performed. A post-hoc z-test on the adjusted residuals with Bonferroni correction was applied to detect between which categories there were significant differences.

The generalized linear model, with an ordered dependent variable and a probit link function, was performed. We initially selected predictors with a p-value <0.20 in the bivariate analysis. Final predictive variables were selected from a backward stepwise procedure; we carried out likelihood ratio test to assess the goodness of fit of the competing statistical models.

All statistical analyses were done with SPSS for Windows version 23.0 and p-values under 0.05 were considered statistically significant.

### Ethical approval

The study was approved by the Balearic Islands Research Ethics Committee (IB3738/18 PI). An information sheet and informed consent was given to the participants before undertaking the interview. Signed agreement of the forms was compulsory to participate.

## Results

Sociodemographic, individual, and labour characteristics of HHs included in the sample (n = 1,043) are shown in Table 1. Most HHs had Spanish nationality, had attained compulsory education, had an external LOC (n = 617/1,035; 59.6% IC95% 56.6–62.6), a recurring seasonal contract, and worked in a hotel (mostly a 4-star hotel).

### Domestic and care burden: Descriptive results

Regarding variables related to household, 5.2% of the participants lived alone; the rest lived in households that were formed, on average, by 3.2 people (±1.1). Forty-six per cent (n = 481) lived with children under 15 years old and 2.3% (n = 24) with people older than 74.

In Table 2, results on the domestic and care burden (multiple answer questions) are displayed. In general terms, most HHs shared the care of children and disabled people with their partner; on the contrary, most HHs pointed out to care themselves alone for the elderly.

**Table 1. Sociodemographic, individual and labour variables.**

|  | n (%) | 95%CI |
|---|---|---|
| Age (n = 1,037) |  |  |
| Under 35 years old | 217 (20,9) | 18,6–23,5 |
| From 35 to 44 years old | 354(34,1) | 31,3–37,1 |
| From 45 to 54 years old | 293(28,3) | 25,6–31,1 |
| 55 years old and over | 173 (16,7) | 14,5–19,1 |
| Nationality (n = 1,043) |  |  |
| Spanish | 563 (54.0) | 50.9–57.0 |
| Double nationality | 182 (17.4) | 15.2–19.9 |
| Foreign | 268 (28.6) | 25.8–31.4 |
| Educational level attained (n = 1,041) |  |  |
| Illiterate/ primary incomplete | 34 (3.3) | 2.3–4.5 |
| Compulsory education (primary and secondary) | 591 (56.8) | 53.7–59.8 |
| Post-compulsory secondary education | 298 (28.6) | 25.9–31.5 |
| University | 118 (11.3) | 9.5–13.4 |
| Type of contract (n = 1,016) |  |  |
| Permanent | 63 (6.2) | 4.8–7.9 |
| Recurring seasonal contract | 551 (54.2) | 51.1–57.3 |
| Temporary | 402 (39.6) | 36.5–42.6 |
| Type of establishment (n = 1,043) |  |  |
| Hotel | 625 (59.9) | 56.9–62.9 |
| Apart-hotel | 297 (28.5) | 25.8–31.3 |
| Rural hotel | 82 (3.7) | 2.7–5.1 |
| Others | 102 (7.9) | 6.3–9.7 |
| Hotel category |  |  |
| 1* | 9 (1.0) | 0.4–1.8 |
| 2* | 37 (4.0) | 2.8–5.4 |
| 3* | 210 (22.5) | 19.9–25.3 |
| 4* | 574 (61.5) | 58.3–64.7 |
| 5* | 103 (11.0) | 9.1–13.2 |
|  |  | SD |
| Age | 43.3 | 42.6–43.9 |
| Years working as hotel housekeeper | 10.7 | 1.9 |
| Months worked last season | 7.0 | 2.0 |
| Hours worked per week | 40.7 | 5.6 |
| Number of rooms/day cleaned | 18.07 | 6.5 |
|  |  | MD |
| **IQR** |  |  |
| Level of stress (min.1- max.7) | 5.0 | 4.0–6.0 |
| Satisfaction with their job (min.1-max.7) | 4.0 | 2.0–6.0 |
| Satisfaction with their wage (min.1-max.7) | 5.0 | 3.0–7.0 |

Note

SD: standard deviation.

MD: median.

IQR: inter quartile range.

CI: confidence interval.

**Table 2. Main person in charge of care and domestic tasks.**

| Main person in charge of. . . | Younger than 15 y.o. (n = 522) | | Disabled person (n = 31) | | Older than 74 y.o. (n = 55) | | Housework (excluding those who live alone) (n = 989) | |
|---|---|---|---|---|---|---|---|---|
| | n (%) | 95%IC | n (%) | 95%IC | n (%) | 95%IC | n (%) | 95%IC |
| HHs alone | 136 (26.1) | 22.3–30.0 | 9 (29.0) | 14.2–48.0 | 43 (78.2) | 65.0–88.2 | 374 (33.3) | 38.0–44.2 |
| Her partner alone | 14 (2.7) | 1.5–4.5 | 1 (3.2) | 0.1–16.7 | 1 (1.8) | 0.0–9.7 | 44 (3.9) | 3.1–5.7 |
| HHs sharing with her partner | 302 (57.9) | 53.5–62.1 | 15 (48.4) | 30.2–66.9 | 6 (10.9) | 4.1–22.2 | 458 (40.8) | 41.6–47.8 |
| HHs sharing with another person who is not her partner | 60 (11.5) | 8.9–14.5 | 3 (9.7) | 2.0–25.8 | 2 (3.6) | 0.4–12.5 | 163 (14.5) | 13.7–18.3 |
| Another person of the household (not her partner) | 41 (7.9) | 5.7–10.5 | 2 (6.5) | 0.8–21.4 | 1 (1.8) | 0.0–9.7 | 67 (6.0) | 5.1–8.2 |
| A hired person | 19 (3.6) | 2.2–5.6 | 1 (3.2) | 0.1–16.7 | - | - | 5 (0.4) | 0.2–1.1 |
| Another person who does not live in the household | 31 (5.9) | 4.1–8.3 | 2 (6.5) | 0.8–21.4 | - | - | 3 (0.3) | 0.1–0.9 |
| Social services | 1 (0.2) | 0.0–1.1 | 2 (6.5) | 0.8–21.4 | - | - | n.a. | - |
| Him/Herself | 38 (7.3) | 5.2–9.9 | 8 (25.8) | 11.9–44.6 | 2 (3.6) | 0.4–12.5 | n.a | - |
| Another situation | 10 (1.9) | 0.9–3.5 | 1 (3.2) | 0.1–16.7 | - | - | 8 (0.7) | 0.3–1.5 |

Regarding housework, there are similar proportions of HHs who assumed the tasks alone and HHs who shared tasks with their partner.

Domestic and family variables, difficulties in balancing work and domestic responsibilities, and the frequency of being overwhelmed by daily tasks are shown in Table 3. Most HHs perceived having normal social support, stated having difficulties making ends meet, and balancing work and care responsibilities. More than a third HHs indicated to be very frequently overwhelmed by daily tasks.

## Difficulties balancing work and family life

Table 4 shows the results of the relationship between individual, sociodemographic, domestic, family and labour variables with WLB. A higher proportion of HHs who reported difficulties in WLB were below 46 years old (p<0.001), lived in households composed by two or more members (p = 0.001), had at least 1 dependant (p<0.001), were the main person in charge or

**Table 3. Domestic and family variables.**

| | n (%) | CI95% |
|---|---|---|
| Social Support (DUKE-UNC) (n = 1,025) | | |
| Low | 65 (6.3) | 4.9–8.0 |
| Normal | 960 (93.7) | 92.0–95.1 |
| Difficulties in making ends meet (n = 1,031) | | |
| Easy | 446 (43.3) | 40.2–46.3 |
| Difficult | 585 (56.7) | 53.7–59.8 |
| Difficulties in WLB (n = 1,035) | | |
| Very easy | 101 (9.8) | 8.0–11.7 |
| Rather easy | 348 (33.6) | 30.7–36.6 |
| Rather difficult | 428 (41.4) | 38.3–44.4 |
| Very difficult | 158 (15.3) | 13.1–17.6 |
| Frequency of being overwhelmed by daily tasks (n = 1,043) | | |
| Very frequently | 365 (35.2) | 32.3–38.2 |
| Sometimes | 426 (41.0) | 38.0–44.1 |
| Almost never | 247 (23.8) | 21.2–26.5 |

**Table 4. Association between sociodemographic, economic and household characteristics, labour variables, domestic burden, locus of control, social support and work-life balance.**

| | Work life balance | | | |
| | Easy | Difficult | | |
| | N (%) | | Chi squared | p-value |
|---|---|---|---|---|
| **Age** (n = 1,028) | | | | |
| <46 y.o. | 224 (37.3) | 376 (62.7) | 20.1 | <0.001 |
| ASR | -4.5 | 4.5 | | |
| ≥46 y.o. | 220 (51.4) | 208 (48.6) | | |
| ASR | 4.5 | -4.5 | | |
| $\bar{x}$ (SD) | 44.6(10.1) | 42.2(10.0) | | <0.001 |
| **Nationality** (n = 1,035) | | | | |
| Spanish | 245 (43.8) | 315 (56.3) | 4.6 | 0.101 |
| ASR | 0.3 | -0.3 | | |
| Double nationality (Spanish and another) | 88 (49.2) | 91 (50.8) | | |
| ASR | 1.7 | -1.7 | | |
| Foreign | 116 (39.2) | 180 (60.8) | | |
| ASR | -1.7 | 1.7 | | |
| **Educational level** (n = 1,033) | | | | |
| Illiterate/primary incomplete | 15 (44.1) | 19 (55.9) | 1.7 | 0.629 |
| ASR | 0.1 | -0.1 | | |
| Compulsory education (primary and secondary) | 246 (41.8) | 342 (58.2) | | |
| ASR | -1.1 | 1.1 | | |
| Post-compulsory secondary edu. | 138 (46.5) | 159 (53.5) | | |
| ASR | 1.3 | -1.3 | | |
| University | 49 (43.0) | 65 (57.0) | | |
| ASR | -0.1 | 0.1 | | |
| **Locus of control** (n = 1,028) | | | | |
| Internal | 201 (48.2) | 216 (51.8) | 6.6 | 0.010 |
| ASR | -2.6 | 2.6 | | |
| External | 245 (40.1) | 366 (59.9) | | |
| ASR | 2.6 | -2.6 | | |
| **Type of contract** (n = 1.008) | | | | |
| Permanent | 32 (50.8) | 31 (49.2) | 1.4 | 0.484 |
| ASR | 1.2 | -1.2 | | |
| Recurring Seasonal | 238 (43.5) | 309 (56.5) | | |
| ASR | -0.1 | 0.1 | | |
| Temporary | 170 (42.7) | 228 (57.3) | | |
| ASR | -0.5 | 0.5 | | |
| **Hotel category** (n = 926) | | | | |
| 1* | 3 (33.3) | 6 (66.7) | 2.1 | 0.713 |
| ASR | -0.6 | 0.6 | | |
| 2* | 15 (41.7) | 21 (58.3) | | |
| ASR | -0.2 | 0.2 | | |
| 3* | 82 (39.4) | 126 (60.6) | | |
| ASR | -1.2 | 1.2 | | |
| 4* | 255 (44.7) | 316 (55.3) | | |
| ASR | 1.3 | -1.3 | | |
| 5* | 43 (42.2) | 59 (57.8) | | |
| ASR | -0.2 | 0.2 | | |
| **Number of household members** (n = 1,035) | | | | |

(*Continued*)

**Table 4.** (Continued)

| | | Work life balance | | | |
| | | Easy | Difficult | | |
| | | N (%) | | Chi squared | p-value |
| | 1 | 31 (55.4)* | 25 (44.6)* | 14.0 | 0.001 |
| | ASR | 1.9 | -1.9 | | |
| | 2 or three | 269 (47.0)* | 303 (53.0)* | | |
| | ASR | 2.6 | -2.6 | | |
| | 4 or more | 149 (36.6)* | 258 (63.4)* | | |
| | ASR | -3.5 | 3.5 | | |
| **Care of dependents (children <15 y.o.,>74y.o. and disabled) (n = 1,035)** | | | | | |
| | Not having any dependent | 304 (50.2) | 302 (49.8) | 27.4 | <0.001 |
| | ASR | 5.2 | -5.2 | | |
| | Having some dependent | 145 (33.8) | 284 (66.2) | | |
| | ASR | -5.2 | 5.2 | | |
| **Domestic tasks (n = 1,009)** | | | | | |
| | HHS is the person in charge of the care | 124 (35.2)* | 228 (64.8)* | 14.8 | 0.001 |
| | ASR | -3.8 | 3.8 | | |
| | She shares the responsibility of the care | 258 (48.0)* | 279 (52.0)* | | |
| | ASR | 3.2 | -3.2 | | |
| | Another person is the main person in charge | 56 (46.7) | 64 (53.3) | | |
| | ASR | 0.8 | -0.8 | | |
| **Social support (n = 1,018)** | | | | | |
| | Low | 14 (22.6) | 48 (77.4) | 11.8 | 0.001 |
| | ASR | -3.4 | 3.4 | | |
| | Normal | 429 (44.9) | 527 (55.1) | | |
| | ASR | 3.4 | -3.4 | | |
| **Making ends meet (n = 1,027)** | | | | | |
| | Easy | 250 (56.3) | 194 (43.7) | 53.6 | <0.001 |
| | ASR | 7.3 | -7.3 | | |
| | Difficult | 195 (33.4) | 388 (66.6) | | |
| | ASR | -7.3 | 7.3 | | |
| | | $\bar{x}$ (SD) | | p-value** | |
| **Months worked last season (n = 1,042)** | | 7.1(2.0) | 6.9 (2.0) | 0.024 | |
| **Number of hours worked/week (n = 1,042)** | | 40.2 (5.0) | 41.1 (5.9) | 0.008 | |
| **Number of rooms cleaned/day (n = 1,043)** | | 17.5(6.7) | 18.5 (6.4) | 0.011 | |
| **Level of stress at work (n = 1,043)** | | 4.4 (1.9) | 5.3 (1.7) | <0.001 | |
| **Level of job satisfaction (n = 1,043)** | | 5.3 (1.7) | 4.6 (1.9) | <0.001 | |
| **Level of wage satisfaction (n = 1,043)** | | 4.5 (1.9) | 3.8 (2.0) | <0.001 | |

Note

ASR: Adjusted standardized residuals.

Bonferroni post-hoc correction.

*Indicates in which categories there are significant differences.

** t-Student test.

**Table 5. Results of ordered probit models predicting work-family conflict.**

|  | Work-life conflict | |
|---|---|---|
|  | **OR (CI 95%)\*** | **p-value** |
| Living alone | Ref. | |
| Living with one or two people | 1.51 (1.08–2.13) | 0.017 |
| Living with three or more people | 1.77 (1.23–2.54) | 0.002 |
| Difficulties making ends meet | 1.43 (1.24–1.65) | <0.001 |
| Having some dependant | 1.42 (1.22–1.66) | <0.001 |
| HHs is the main responsible of domestic tasks | 1.40 (1.20–1.63) | <0.001 |
| External locus of control | 1.17 (1.02–1.35) | 0.027 |
| Level of stress at work (min 1- max 7) | 1.12 (1.07–1.16) | <0.001 |
| Hours worked/week | 1.02 (1.00–1.03) | 0.013 |
| Satisfaction with their job (min 1- max 7) | 0.94 (0.90–0.98) | 0.003 |
| Satisfaction with their wage (min 1- max 7) | 0.96 (0.92–1.00) | 0.049 |
| Age (years) | 0.99 (0.98–1.00) | 0.004 |

share the responsibility of domestic tasks (p = 0.001), had an external LOC (p = 0.010), had low social support (p = 0.001) and reported difficulties making ends meet (p<0.001).

Regarding labour variables, participants having difficulties in WLB worked on average 6.9 months (p = 0.024), worked 41.1 hours/week (p = 0.008) and cleaned 18.5 rooms/day (p = 0.011). HHs presenting difficulties in combining work and care responsibilities reported higher stress levels and lower scores in job and wage satisfaction (p<0.001).

## Risk and protective factors related to work-life balance

Table 5 shows the results of the ordered probit model for difficulties in WLB. An increased risk of WFC was associated with living with one or more people, having difficulties making ends meet, taking care of any dependant, being the main person in charge of domestic tasks, having an external locus of control, higher levels of stress, working more hours a week and being younger. Factors that reduce the risk of WFC are satisfaction with their job and with their wage.

## Discussion

The interest on the articulation of paid and unpaid work has been increasing throughout the past years. Despite the growth of the participation of women in paid work and the labour market, they are still the main person in charge of the household for unpaid care and housework. Conflict between work and non-work spheres has been widely explored due to its negative outcomes on life satisfaction and well-being. However, few studies have researched this phenomenon among working class or unskilled workers. The objectives of this study were to describe the domestic burden of HHs and to assess personal, labour, economic, and family factors that explain WFC within HHs, a non-skilled 100% feminized occupational group in the Balearic Islands (Spain).

Our findings revealed that HHs present difficulties for combining work and family life to a great extent. The proportion of HHs reporting to have difficulties in the articulation of paid work and other responsibilities (56.7%) is higher than that of Spanish working women (51.9%) according to the European Quality of Life Survey 2016 [6]. Characteristics of HHs' job–inflexible work hours, high physical demands, etc.- might represent a barrier to attend personal and family demands. Additionally, the frequency of being overwhelmed by daily tasks is higher

among HHs (76.2%) compared to working women according to the last available data from the Spanish Time Use Survey (2002–03) [48], in which 61.6% declared to be overwhelmed by daily tasks (16.7% very frequently, and 44.9% sometimes). Part of these differences in comparison to national and international surveys might be explained by the inclusion of a more heterogeneous sample regarding social class, while our study focused on a single occupational group.

Furthermore, these data are consistent with surveys and research on time-use, which have largely reported differences between men and women as well as an unequal involvement in household tasks, this being higher among women [30, 53, 54]. Additionally, and consistent with our results, difficulties in WLB have been reported as being greater among blue collar workers [6].

Ordered probit models allowed us to control for the effect of different variables from both spheres–family/personal and work-. Results regarding factors explaining WFC showed that family demands and its characteristics (having dependants, living with one or more people, being the main person in charge of domestic tasks), having an external LOC, and economic factors (difficulties making ends meet) along with labour aspects (higher levels of stress at work, working more hours) and being younger had a statistically significant association with an increase in WFC. On the contrary, higher levels of job and wage satisfaction had a protective effect on WFC. These relationships show how labour and domestic spheres are non-separate worlds, since the characteristics of one influence the performance of the other. Results also show that WLB is beyond individual willing and is strongly influenced by economic, labour and domestic factors along with gendered allocation of care and housework.

## Domestic sphere and WFC

Regarding variables related to domestic and family sphere, results indicated that there was a higher proportion of HHs who presented WFC among those who cohabited with one or more people, had dependants or were the main responsible person for domestic tasks. The proportion of HHs who assumed alone the care of minor and disabled and household chores was lower than those registered by working women–regardless of their social class- in the Spanish National Health Survey (2011–12)–last data available-[49]. The positive relationship between number of household members and WFC pointed at the fact that HHs did not have enough resources to face the increase in care and housework demands as the number of members increase. Two explanations might account for this relationship. First, it might be explained by the fact that the more members a household has, the more domestic chores have to be done; thus, it might reflect that the increase in housework was probably assumed mainly by HHs, since women have been described to be the main responsible person for care [53, 55]. Second, it appeared that household members were more a burden -because they were dependent (children, older, disabled) and were not able to carry out domestic chores- than a resource for performing domestic tasks. Specially, having children has been associated to an increase in unpaid work [53, 55, 56] and to increased WFC [57].

The distinction between domestic tasks and care work is important because both imply different requirements [1]; housework tends to be routine and sometimes boring, while childcare tends to be perceived positively by parents and as an investment. Moreover, care for dependents cannot be postponed and it must be done on a daily basis. Due to these differences, including these variables separately in the analysis should be assessed as strength of this study.

The positive relationship between assuming more care responsibilities or household chores and WFC can be explained by the time-based conflict, defined as the "time spent on activities

within one role generally cannot be devoted to activities within another role" [11]. As well, energy devoted to one domain cannot be devoted to the other domain. Thus, in the case of HHs, the combination of a high physically demanding and stressful job with a high care and domestic burden probably results in WFC. Additionally, results may reflect the gendered allocation of care and household labour [53] by which women are supposed to be the main responsible person for these demands. Despite this, some studies show that women and men report similar levels of WFC [37, 57], in part because women tend to adjust their working demands (i.e. schedule) in order to respond more adequately to family demands and reduce levels of WFC [28].

Although we did not ask for HHs' satisfaction with working schedule in this study, in a previous qualitative study [19], HHs valued very positively the continuous working day because it was perceived as a facilitator of WLB when they had children; a split schedule would probably make WLB more difficult. Since most HHs worked 40 hours per week, a question about the satisfaction with working schedule would have provided relevant information associated to WFC.

Our findings support that LOC has a negative relationship with WFC: there are lower levels of WFC among people with an internal LOC [58]. People with an internal LOC perceive more control about what happens in their life, so they might be able to better handle stressors. Also, our results pointed out that personality traits might play a role in determining WFC, along with situational and structural factors.

Furthermore, having difficulties making ends meet was an important variable to explain WFC. This is in line with studies which found that higher income levels were associated to lower levels of WFC [28], because workers with lower levels of income have less economic resources to access paid aid to reduce household chores. Despite this, there is mixed evidence regarding the relationship between income level and WFC [16, 59, 60]. Our results seem to reveal that having difficulties making ends meet make feel HHs more pressured to fulfil work demands due to the importance of their wage for the subsistence and well-being of their family. Besides, it may indicate the impossibility to reduce working hours. These results point out to the relevance of social class in explaining different experiences in WFC.

## Labour variables and WFC

Regarding labour variables, our findings showed the importance of the levels of stress [57] in predicting WFC. This is consistent with the strain-based conflict, which "exists when strain in one role affects one's performance in another role" [11]. Thus, tiredness and/or distress produced at work may hinder facing properly the demands from the domestic domain. In the hospitality context, Zhao et al. (2014) [16] concluded that interferences between job and non-work spheres as well as job characteristics increased employees' stress. HHs' job is characterized by inflexible work hours and low control among other factors which may explain the high levels of stress related to WFC. Generally, this evidence manifests that the organization and characteristics of work settings have a straight impact on personal life, and supports the idea about the close interrelationship between work and non-work spheres.

Other labour variables included in the model were wage and job satisfaction. HHs reporting more difficulties to combine work and family demands showed low levels of job and wage satisfaction, a relationship that might be bidirectional. Research has established that experiencing role conflict in life domains entails dissatisfaction and outcomes related to stress as well as consequences over work and non-work spheres [12, 61]. As research indicates, perceiving WFC is likely to imply dissatisfaction across life domains because individuals do not feel able of performing their responsibilities properly.

Our results support the relevance of structural factors and power relations to explain WLB experiences; thus, employees with higher educational level and in managerial positions report lower WFC [62]. Our findings underscore the importance of the intersectionality approach to WFC issues, since different intersected variables are crucial in explaining this social phenomenon. In this sense, according to prior evidence presented in this manuscript, gender and social class interact in their effects on the way individuals experience WFC.

One of the study's limitation is that the data were collected in winter (low season in the Balearic Islands) in order to be able to reach HHs for interviews. Most HHs do not work in winter, which might lead to an under estimation of WLB difficulties when asked for a situation that took place a few weeks or months ago. Second, despite the importance of gender in approaching the work-life interface, comparison with men could not be undertaken because HHs is a 100% feminized occupational group in the sample's context. Third, data collection took place before COVID19 pandemic; thus, the picture presented here has probably changed.

## Future research directions

The interconnection between life and work domains emphasized by the results of our study entails the necessity to incorporate individual, family, labour and economic variables when approaching the work-family interface. Due to the complexity of the WFC phenomenon, adopting an intersectional approach is essential in future studies. This implies including not only gender, but also social class, due to the importance of economic resources at explaining WFC.

## Practical implications

The results presented in this manuscript allow outlining some strategies to improve the articulation of work and life domains among HHs. At work, hospitality companies should implement measures to reduce daily stress, such as diminishing the number of rooms cleaned per day or the variety of tasks carried out by HHs [19]. This kind of measures would increase HHs' autonomy and reduce time pressure; thus stress at work would be reduced. Second, given the relevance of the economic factor in explaining WFC, hospitality companies should improve wage conditions too. Third, public authorities should contribute to a better articulation of work and non-work domains. On one hand, they might launch campaigns to foster co-responsibility between partners for caring and performing household tasks, giving value to reproductive tasks and oriented, above all, towards men. On the other hand, services to support caring (for dependents, but also children) must be fostered, improved, and oriented, above all, towards people with less economic and support resources. This should include affordable childcare services, summer-camps, day-care centres for elderly and home assistance, among other.

## Conclusion

This study provides a detailed understanding of individual, labour, economic and family factors explaining WFC among HHs, a non-skilled 100% feminized occupational group. Risk factors of perceiving difficulties in WLB were living with one or two more people, having difficulties making ends meet, having some dependant, being the main person in charge of domestic tasks, having an external LOC, presenting higher levels of stress at work and being younger. Protective factors from experiencing WFC were satisfaction with their job and with their wage. Thus, our findings highlight the connection between work and personal demands and resources. Additionally, results indicate that intersection between social class and gender, at least, plays a central role in explaining WFC.

## Author Contributions

**Conceptualization:** Xenia Chela-Alvarez, Oana Bulilete, Joan Llobera.

**Data curation:** Xenia Chela-Alvarez.

**Formal analysis:** Xenia Chela-Alvarez, Oana Bulilete, Joan Llobera.

**Investigation:** Xenia Chela-Alvarez.

**Methodology:** Xenia Chela-Alvarez, Joan Llobera.

**Resources:** Joan Llobera.

**Supervision:** M. Esther Garcia-Buades, Victoria A. Ferrer-Perez.

**Writing – original draft:** Xenia Chela-Alvarez.

**Writing – review & editing:** M. Esther Garcia-Buades, Victoria A. Ferrer-Perez, Oana Bulilete, Joan Llobera.

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
