## [Decision Letter · Decision Letter 0]

23 Nov 2022

PONE-D-22-13784Work-family conflict among hotel housekeepers in the Balearic Islands (Spain)PLOS ONE

Dear Chela,

Thank you for submitting your manuscript to PLOS ONE. After careful consideration, we feel that it has merit but does not fully meet PLOS ONE’s publication criteria as it currently stands. Therefore, we invite you to submit a revised version of the manuscript that addresses the points raised during the review process. Do address all the revisions suggested by the reviewers and the editor. 

We look forward to receiving your revised manuscript.

Kind regards,

Sandra Boatemaa Kushitor, Ph.D.

Academic Editor

PLOS ONE

Journal Requirements:

Additional Editor Comments (if provided):

Dear Author,

This is a good paper and makes a significant contribution to understanding work life balance or conflict.

Please use formal language. It will be good to have a language editor review the work. Change most of the very old references. Considering looking at the migration status of the respondents. You can use this demographic characteristics to explain the social support resources available to them.

Reviewers' comments:

Reviewer's Responses to Questions

**Comments to the Author**

1. Is the manuscript technically sound, and do the data support the conclusions?

Reviewer #1: Yes

Reviewer #2: Yes

2. Has the statistical analysis been performed appropriately and rigorously? 

Reviewer #1: Yes

Reviewer #2: Yes

3. Have the authors made all data underlying the findings in their manuscript fully available?

Reviewer #1: Yes

Reviewer #2: Yes

4. Is the manuscript presented in an intelligible fashion and written in standard English?

Reviewer #1: Yes

Reviewer #2: Yes

5. Review Comments to the Author

Reviewer #1: Major Issues

1. WFC variable has five categories. Therefore, instead of using just binary logistic regressions, authors must use ordered probit models.

Minor Issues

1. In the conclusion, authors wrote that HHs is a 100% feminized occupational group. However, they must mention that their sample do not include men in the data collection part (even in the introduction).

2. Proofreading is required; for example, city names must be written in English e.g., Ibiza.

Reviewer #2: Theoretical introduction: Well written, with a good description of work-family conflict theory and very good interconnection between theory and the population in analysis. Interesting interpretation of the context (profession and country). In general, the introduction itself is well developed, with a good description of the study goal. Could be interesting some more theoretical development about work-family balance, not specifically as an absence of WFC, but as a concept in itself.

Methodology: Good description of measures evaluation and methodological concepts used. Very positive number of participants, with diversity of sociodemographic variables.

Results: Well written, with a good concise presentation of tables and a logical order. Good description of results in all variables which gives a suitable overview of the sample.

Discussion: Important interconnection with theoretical framework presented throughout the manuscript. Important theoretical implications for social class and work-family conflict, extending studies regarding higher educational level positions.

Bibliography: Well-chosen, with a good range of diversity of fonts (articles, websites, national reports). Thematic of work-family conflict pertinently sustained.

General commentary: Thank you for the opportunity of reading this manuscript. It’s an important topic with an interesting, and not so-common, population on study. Important to complement and extend work-family research, specifically with unskilled population and beyond organizational 9 to 5 models.

Considering all the mentioned above, I suggest an acceptation of the manuscript, with minor improvement (e.g., work-life balance theoretical development).

6. PLOS authors have the option to publish the peer review history of their article (what does this mean?). If published, this will include your full peer review and any attached files.

Reviewer #1: No

Reviewer #2: **Yes: **Maria José Chambel

---

## [Author Response · Author response to Decision Letter 0]

6 Jan 2023

Please, find below the responses to editor and reviewers: 

i.Please use formal language. It will be good to have a language editor review the work.

Language has been reviewed and corrections have been made. Please, see the "response to reviewers PONE-D-22-13784" document attached. 

ii. Change most of the very old references. 

Most the old references has been replaced. Please, see the "response to reviewers PONE-D-22-13784" document attached.

iii. Considering looking at the migration status of the respondents. You can use this demographic characteristics to explain the social support resources available to them.

We really appreciate this suggestion. We performed the ordered probit models as suggested by Reviewer#1 and social support was not statistically significant. Please, see the "response to reviewers PONE-D-22-13784" document attached.

Reviewer #1

MAJOR ISSUES

1. WFC variable has five categories. Therefore, instead of using just binary logistic regressions, authors must use ordered probit models.

We really appreciate this comment. We performed ordered probit models with the WFC variable (with four categories, because the fifth category [do not know/do not answer] was labeled as missing). Please, see the "response to reviewers PONE-D-22-13784" document attached.

MINOR ISSUES

1. In the conclusion, authors wrote that HHs is a 100% feminized occupational group. However, they must mention that their sample do not include men in the data collection part (even in the introduction).

Thank you for your observation. We included this information. Please, see the "response to reviewers PONE-D-22-13784" document attached.

2. Proofreading is required; for example, city names must be written in English e.g., Ibiza.

Thank you for your comment. “Eivissa” has been changed for “Ibiza”.

Reviewer #2

Theoretical introduction: Could be interesting some more theoretical development about work-family balance, not specifically as an absence of WFC, but as a concept in itself. Please, see the "response to reviewers PONE-D-22-13784" document attached.

---

## [Decision Letter · Decision Letter 1]

14 Feb 2023

Work-family conflict among hotel housekeepers in the Balearic Islands (Spain)

PONE-D-22-13784R1

Dear Dr. Chela,

We’re pleased to inform you that your manuscript has been judged scientifically suitable for publication and will be formally accepted for publication once it meets all outstanding technical requirements.

Kind regards,

Sandra Boatemaa Kushitor, Ph.D.

Academic Editor

PLOS ONE

Additional Editor Comments (optional):

Reviewers' comments:

Reviewer's Responses to Questions

**Comments to the Author**

1. If the authors have adequately addressed your comments raised in a previous round of review and you feel that this manuscript is now acceptable for publication, you may indicate that here to bypass the “Comments to the Author” section, enter your conflict of interest statement in the “Confidential to Editor” section, and submit your "Accept" recommendation.

Reviewer #1: All comments have been addressed

2. Is the manuscript technically sound, and do the data support the conclusions?

Reviewer #1: Yes

3. Has the statistical analysis been performed appropriately and rigorously? 

Reviewer #1: Yes

4. Have the authors made all data underlying the findings in their manuscript fully available?

Reviewer #1: (No Response)

5. Is the manuscript presented in an intelligible fashion and written in standard English?

Reviewer #1: (No Response)

6. Review Comments to the Author

Reviewer #1: (No Response)

7. PLOS authors have the option to publish the peer review history of their article (what does this mean?). If published, this will include your full peer review and any attached files.

Reviewer #1: No

---

## [Editor Report · Acceptance letter]

17 Feb 2023

PONE-D-22-13784R1 

Work-Family conflict among hotel housekeepers in the Balearic Islands (Spain). 

Dear Dr. Chela-Alvarez:

I'm pleased to inform you that your manuscript has been deemed suitable for publication in PLOS ONE. Congratulations! Your manuscript is now with our production department. 

Kind regards, 

on behalf of

Dr. Sandra Boatemaa Kushitor 

Academic Editor

PLOS ONE